# Super-Resolution Fluorescence Microscopy Reveals Clustering Behaviour of *Chlamydia pneumoniae’s* Major Outer Membrane Protein

**DOI:** 10.3390/biology9100344

**Published:** 2020-10-20

**Authors:** Amy E. Danson, Alex McStea, Lin Wang, Alice Y. Pollitt, Marisa L. Martin-Fernandez, Isabel Moraes, Martin A. Walsh, Sheila MacIntyre, Kimberly A. Watson

**Affiliations:** 1School of Biological Sciences, University of Reading, Berkshire RG6 6AS, UK; a.pollitt@reading.ac.uk (A.Y.P.); s.macintyre@reading.ac.uk (S.M.); 2Diamond Light Source, Harwell Science and Innovation Campus, Oxfordshire OX11 0DE, UK; martin.walsh@diamond.ac.uk; 3Research Complex at Harwell, Harwell Science and Innovation Campus, Oxfordshire OX11 0FA, UK; isabel.moraes@npl.co.uk; 4Central Laser Facility, Research Complex at Harwell, Science and Technology Facilities Council, Rutherford Appleton Laboratory, Harwell, Didcot, Oxford OX11 0QX, UK; aem39@bath.ac.uk (A.M.); lin.wang@stfc.ac.uk (L.W.); marisa.martin-fernandez@stfc.ac.uk (M.L.M.-F.); 5National Physical Laboratory, Teddington TW11 0LW, UK

**Keywords:** membrane proteins, *Chlamydia pneumoniae*, fluorescence microscopy, bacterial structures

## Abstract

**Simple Summary:**

*Chlamydia* is an infamous sexually transmitted bacterium that also has a less well-known role in human respiratory infections, which has evolved a unique cell structure to enable its survival within the body. Covering the surface of this infectious cell is a strong mesh-like network made up of many different proteins which protects the cell against damage. This research focussed on the most abundant protein within this mesh, the Major Outer Membrane Protein (MOMP), and introduced a series of mutations designed to prevent the mesh from forming completely. The effect of the mutations was visualised by adding a bright fluorescent dye to each MOMP, which was then examined with a high-resolution fluorescence microscope capable of showing us each individual cell and the MOMPs at their surface. With statistical analysis, we observed that certain mutations disrupted the connections between MOMPs, giving us greater insight into how *Chlamydia* forms these interactions. *Chlamydia* is an extremely prevalent disease amongst the global population, and whilst treatable, there is currently no available vaccine. By researching *Chlamydia’s* biology and its method of evading our immune system, we can not only further our understanding of this complex bacterium, but also develop novel therapeutics for its treatment and prevention.

**Abstract:**

*Chlamydia pneumoniae* is a Gram-negative bacterium responsible for a number of human respiratory diseases and linked to some chronic inflammatory diseases. The major outer membrane protein (MOMP) of *Chlamydia* is a conserved immunologically dominant protein located in the outer membrane, which, together with its surface exposure and abundance, has led to MOMP being the main focus for vaccine and antimicrobial studies in recent decades. MOMP has a major role in the chlamydial outer membrane complex through the formation of intermolecular disulphide bonds, although the exact interactions formed are currently unknown. Here, it is proposed that due to the large number of cysteines available for disulphide bonding, interactions occur between cysteine-rich pockets as opposed to individual residues. Such pockets were identified using a MOMP homology model with a supporting low-resolution (~4 Å) crystal structure. The localisation of MOMP in the *E. coli* membrane was assessed using direct stochastic optical reconstruction microscopy (dSTORM), which showed a decrease in membrane clustering with cysteine-rich regions containing two mutations. These results indicate that disulphide bond formation was not disrupted by single mutants located in the cysteine-dense regions and was instead compensated by neighbouring cysteines within the pocket in support of this cysteine-rich pocket hypothesis.

## 1. Introduction

*Chlamydia* is a genus of Gram-negative obligate intracellular bacteria, infamous for its sexually transmitted infection in humans, propagated by the strain *Chlamydia trachomatis*. Within this genus are two additional strains, *Chlamydia pneumoniae*, a human respiratory agent, and *Chlamydia psittaci*, a zoonotic pathogen endemic within the avian population. *C. pneumoniae* is responsible for 10% of community-acquired pneumonia in humans [1] as well as other respiratory diseases such as sinusitis and bronchitis. *C. pneumoniae* also has an often-debated role in inflammatory diseases such as atherosclerosis [2,3,4], reactive arthritis [5,6,7] and asthma [8,9]. The majority of *C. pneumoniae* infections occur from early childhood through to teenage years, leading to a high prevalence within the population, highlighting the importance of early intervention for adequate control in reducing *Chlamydia* prevalence, in addition to the associated diseases.

Unique to *Chlamydia* is its biphasic lifecycle consisting of infectious elementary bodies (EBs) that circulate the host, and upon infection of a cell, differentiate into a larger and metabolically active form known as reticulate bodies (RBs). Typically, Gram-negative bacteria consist of an inner and outer membrane separated by the periplasmic space in which the sugar polymer peptidoglycan exists to provide cell rigidity and prevent osmotic lysis. However, in *Chlamydia*, the peptidoglycan layer is absent for much of the lifecycle and has only been detected at the septum of dividing RBs [10]. Instead, it is the chlamydial outer membrane complex (COMC), previously referred to as the P-layer [11,12,13,14], that fulfils the role of providing much needed structural rigidity in circulating EBs. The essential proteins, which consist of 17 cysteine-rich outer membrane proteins (in order of abundance: MOMP, OmcB, PmpG, PmpH, PmpE, PulD/YscC, OprB, CTL0887, PorB, OmcA, PmpB, PmpC, PmpF, CTL0541, OMP85, CTL0645 and Pal) [15], interlink through intermolecular disulphide bonds to form a compensatory protein mesh within the EB outer membrane. The major outer membrane protein (MOMP) is a key contributor to this mechanism, accounting for ~60% of the COMC protein [16], and is conserved amongst chlamydial species with between 63 and 75% identity. *C. pneumoniae* MOMP is a β-barrel protein containing nine cysteine residues, accounting for 2.4% of the structure, of which seven are conserved across the species [17]. Due to its abundance and high cysteine-rich content, it is highly likely that MOMP forms a number of intermolecular disulphide bonds with other MOMPs in the outer membrane. The interlinking of MOMP with some of the less abundant COMC proteins listed above, such as PorB and the polymorphic membrane proteins (Pmps), may also be required to form a complete mesh framework, although this remains to be investigated. Aside from the two published homology models for MOMP in *C. pneumoniae* and *C. trachomatis*, respectively [18,19], only one other COMC protein homology model exists for PorB [20], highlighting the current difficulties in producing reliable models for these proteins.

Previous experimental research has been conducted on *C. trachomatis* MOMP in an effort to identify the cysteine residues likely to be involved in disulphide bonding; however, these reports appear to be conflicting. One group predicted that cysteine pairs C48–C55 (corresponding to C49–C56 in *C. pneumoniae* MOMP) and C201–C203 form intramolecular disulphide bonds in refolded MOMP, leaving the remaining cysteine residues free and available for intermolecular disulphide bonding [21]. Further studies also investigating *C. trachomatis* MOMP inferred that C26–C337 and C116–C208 (corresponding to C49–C353 and C136–C226 in *C. pneumoniae* MOMP, respectively) formed intramolecular disulphide bonds [22]. Based on steric hindrance, it seems highly unlikely that residues C201–C203 would form an intramolecular disulphide bond due to their close proximity and likely positioning on the same β-strand, which would not serve to enhance the stability of the barrel and, therefore, form a redundant interaction. A mutant recombinant *C. trachomatis* MOMP containing alanine substitutions for all nine cysteines residues was also demonstrated to be correctly folded and inserted into the *E. coli* outer membrane by gel exclusion experiments of the outer membrane preparation (solubilised in 1% LDAO), suggesting that these predicted cysteine pairs may not in fact be forming intramolecular disulphide bonds, which are typically required for correct protein folding and stability [23]. Additionally, MOMP was also shown to be inserted into the outer membrane in its reduced thiol state upon differentiation in the EB, indicating that disulphide bonding occurs during COMC formation [24].

A review into *Chlamydia* disulphide bonding by Christensen et al. offers a comprehensive analysis of the role of cysteines in *Chlamydia* and the COMC [17], although a mechanism of how oxidation occurs (via these cysteine-rich proteins) is currently unknown. However, a number of putative oxidoreductases of the Dsb family have been proposed, namely DsbA, DsbB and DsbD, which have approximately 20% sequence similarity to those found in *E. coli* [17], whereby DsbA and DsbB were later confirmed experimentally as a redox pair in *Chlamydia trachomatis* [25]. Dsb mutants in *E. coli* have also been shown to be viable under aerobic conditions, despite the reliance of two essential proteins, LptD and FtsN, on disulphide bonding for activity, suggesting that the presence of oxygen alone is sufficient for disulphide bond formation in *E. coli*, although at a reduced efficiency [26].

Due to the small size of bacterial cells, for example of around just 2 μm in length for *Escherichia coli*, the analysis of cell components and structures by light microscopy methods has largely been limited by their resolving power. With total internal reflection fluorescence (TIRF) microscopy, however, it has been possible to characterise the properties of membrane-associated proteins in bacterial cells for both *Bacillus subtilis* [27] and *E. coli* [28], although literature dedicated to detailed and comprehensive bacterial cell sample preparation on other species is still insufficient. In this work, we used epifluorescence and TIRF microscopy to obtain qualitative data to observe the variations in recombinant MOMP (rMOMP) membrane localisation. Furthermore, in order to acquire higher-resolution images and obtain quantitative data for clustering analysis, direct stochastic optical reconstruction microscopy (dSTORM) [29] was used. In conventional epifluorescence microscopy, it is impossible to localise a large number of fluorophores within close proximity below the diffraction limit, which results in low-resolution images. dSTORM is a fluorescent microscopy method based upon single-molecule localisation, whereby the use of photoswitchable organic fluorophores for labelling is imperative [30]. These fluorophores can be controlled with exposure to light and a reducing agent, creating an on/off cycling state known as ‘blinking’, where at any given time, only a sparse subset of fluorophores are in the on state and, therefore, not overlapping. The collection of a large number of frames, for example over 20,000, captures the localisation of each fluorophore within the sample for which the centroid positions can be calculated from Gaussian fit to the point spread function (PSF) and reconstructed into a super-resolution image. As a result, the resolution of dSTORM is not limited by diffraction but rather the measured localisation precision [31] and labelling density [32] and is reported to be as good as 20 nm [33]. It is not uncommon in dSTORM imaging for the same fluorophore to be present in a number of frames due to the on/off cycling times. Therefore, in postimaging analysis, such data (known as trails) can be grouped according to the fluorophore from calibrated on/off times and known capture radii. Concurrent with STORM’s development [33], other research groups established photoactivated localisation microscopy (PALM) [34] and fluorescence photoactivated localisation microscopy (FPALM) [35], all three of which adhere to the same principle, although the term dSTORM will be used herein.

Owing to the high number of cysteine residues in MOMP, it is hypothesised that recombinant expression in *E. coli* results in ‘clustering’ within the membrane due to intermolecular disulphide bonding, exhibited as spotted fluorescence. Light microscopy experiments by Findlay et al. which fluorescently labelled rMOMP expressed in *E. coli* to demonstrate membrane localisation exhibited this uneven membrane distribution [23]; thus, it is hypothesised that the neutralisation of particular cysteine residues believed to hold dominant roles in COMC intermolecular disulphide bonding, through mutation to alanine, would likely decrease MOMP clustering and increase dispersion within the membrane. Currently, there is a paucity of data on the interactions that occur between cysteine-rich proteins within the COMC. Being the most abundant outer membrane protein, it is likely that MOMP plays an extremely substantial role in the COMC architecture and likely forms a number of intermolecular disulphide bonds with other MOMPs. By developing a more detailed molecular understanding of the COMC, which interactions occur and how they are initiated, progress can be made towards the design and development of robust vaccines and antimicrobials, which would likely be particularly effective against the COMC due to its external exposure on the circulating EBs, as well as it playing a major role in cell structural stability.

## 2. Materials and Methods

All chemicals and reagents were acquired from Sigma-Aldrich unless otherwise stated.

### 2.1. Construct Design

*Chlamydia pneumoniae* strain AR39 wild-type MOMP was cloned into the pET101/D-TOPO vector, which encodes a C-terminal His-tag. Cysteine mutants (C136A, C201A, C203A, C201/203A, C136/201A, C136/203A, C226A) were created using the QuikChange XL site-directed mutagenesis kit (Agilent). Primers were purchased from Eurofins MWG and are supplied in Appendix A.

### 2.2. Bacterial Growth Conditions and Induction of rMOMP

*E. coli* strain C41 (DE3) was transformed with pET101/D-TOPO-MOMP or derived mutants thereof and grown on LB agar plates containing 0.6% (*w*/*v*) glucose and 100 µg mL^−1^ ampicillin at 37 °C. A single colony was inoculated into 10 mL of fresh LB media containing both glucose and ampicillin and grown overnight at 30 °C with shaking at 225 rpm. Overnight cultures were diluted 100-fold into fresh media supplemented with an antibiotic and grown at 37 °C with shaking at 225 rpm until an OD_600_ of ~0.5 was reached. Cells were induced with 1 mM IPTG and grown at a reduced temperature of 25 °C for an additional 3 h. At the time of harvesting, cells were resuspended to an OD_600_ of 0.5 (equating to approximately 4 × 10^8^ cells mL^−1^) and pelleted for 3 min at 3500× *g* before resuspension in phosphate-buffered saline (PBS). Membranes for Western blot analysis were isolated from 1 L cultures with an additional centrifugation step at 16,740× *g* following cell lysis with a cell disruptor (Constant Systems) in order to remove inclusion bodies before membranes were pelleted at 195,462× *g* for 2 h. Inner membranes were then solubilised in solubilisation buffer (20 mM Tris pH 7.5, 300 mM NaCl, 1 mM DTT, 10% glycerol) containing 2% sarkosyl [36] for 1 h at 4 °C before centrifugation at 195,462× *g* for 45 min. Outer membranes in the insoluble pellet were resuspended in solubilisation buffer and supplemented with 1% SB3-14 and stirred for 2 h at 4 °C, followed by another ultracentrifugation step. The supernatant containing detergent soluble outer membranes was retained and used for Western blotting analysis.

### 2.3. Slide Preparation

An Ibidi 8-well glass-bottom µ-slide (170 µm ± 5 µm) was incubated with 0.01% poly-L-lysine for 10 min then washed three times with PBS. Bacterial cells, resuspended in PBS, were immobilised on the slides for 1 h, followed by additional washing with PBS. Cells were fixed with 2% (*v*/*v*) formaldehyde-PBS for a further 10 min before being washed three times with PBS. The slide was stored in PBS at 4 °C until the day of imaging.

For immunostaining, storage PBS was aspirated off and the slide was incubated in 50 mM ammonium chloride for 10 min in order to quench any residual fixative. After washing with PBS, cells were permeabilised with 0.1% (*v*/*v*) Triton X-100 for 5 min, followed again by washing. Cells were then incubated for 45 min with 100 µg mL^−1^ lysozyme and washed with PBS. The slide was blocked for 1 h in 3% (*w*/*v*) BSA-PBS blocking buffer before incubation with a primary antibody or conjugated antibody in 3% (*w*/*v*) BSA-PBS for 1 h. Wells were washed and incubated for 1 h with a secondary antibody. A set of control cells, not expressing rMOMP, were not immunostained to assess for bacterial autofluorescence. Wells were washed and stored in fresh PBS for imaging.

### 2.4. Epifluorescence and TIRF Imaging Conditions

Samples were imaged by TIRF and epifluorescence using a Nikon Eclipse Ti inverted microscope with a 100× oil immersion objective and 1.49X NA TIRF objective. rMOMP-expressing cells were stained using an anti-His rabbit primary antibody coupled with an anti-rabbit Alexa Fluor 488 secondary antibody. Control *E. coli* cells were stained using an anti-OmpA rabbit primary antibody also coupled with the anti-rabbit Alexa Fluor 488 secondary antibody. Fluorophores were excited using the 488 nm laser. Images were captured with an Andor camera after a 20–200 ms exposure time and visualised using the NIS-Elements AR 4.5 imaging software.

### 2.5. dSTORM Immunostaining Procedure

The immunostaining procedure for dSTORM closely resembles that of epifluorescence and TIRF microscopy imaging, although with a few minor differences. In dSTORM experiments, a single-step labelling method using Alexa Fluor 647 conjugated anti-His in 3% (*w*/*v*) BSA-PBS was utilised for rMOMP samples. For OmpA control samples, an anti-OmpA rabbit primary antibody was used as before instead and coupled to an Alexa Fluor 647 secondary antibody, also in 3% (*w*/*v*) BSA-PBS.

### 2.6. dSTORM Imaging Conditions

dSTORM experiments were carried out using the Octopus facility at the Central Laser Facility (CLF), Harwell Campus, UK. Data were collected using a Zeiss Elyra PS.1 microscope fitted with an alpha Plan-Apochromat 100×/1.46 oil DIC M27 objective lens using Immersol 518 F immersion oil (Zeiss). Samples were imaged in 0.1 M DTT photoswitching buffer as opposed to PBS. Fluorophores were excited using the 642 nm laser, raised to 0.55–1.1 kW/cm^2^ to achieve a stable blinking state, before being lowered to 0.28 kW/cm^2^ for data collection and detected with a LBF 561/642 dual-band dichroic filter. Twenty thousand images in the field of view of 12.8 × 12.8 µm were recorded with an exposure time of 20 ms and a camera gain of 300 using an EMCCD camera (Andor iXon DU 897). The axial drift of the samples was corrected every 500 frames in real time using definite focus functionality in the microscope.

### 2.7. Clustering Analysis of dSTORM Images

The collected data were processed in Zen Black 2012 software (Zeiss) using the PALM module where the peak mask size was set as 9 pixels and the peak intensity to noise ratio was set as 6 to reject abnormally or dimly emitting fluorophores. Overlap of molecules was accounted for through a multiobject fitting algorithm using a PSF half-width of 177.9 nm. A model-based drift correction using automatic segmentation no bigger than 8 was applied, and trails were grouped for Alexa Fluor 647 as follows: max on time of 5 frames, max off gap of 10 frames and a capture radius of 2 pixels, based on previously calibrated data [37,38]. The ASCII text files detailing spatial localisation information for each molecule were used in clustering analysis with ClusDoC software [38]. Using the ClusDoC GUI, ROIs were manually defined by the user; in this instance, as whole *E. coli* cells. The density-based spatial clustering of applications with noise (DBSCAN) algorithm was implemented on the ROIs using the following parameters: epsilon—20 nm; min points—3; plot cut off—10; threads—2; L (r)—r—50 nm; smooth radius—14 nm.

## 3. Results

### 3.1. Cysteine Mutant Design

A number of cysteine mutants (C136A, C201A, C203A, C201/203A, C136/201A, C136/203A, C226A) were designed based on the cysteine-rich pocket hypothesis proposed herein. Due to the high number of cysteine-rich proteins contributing to the COMC, which between them contain 194 cysteine residues, it would likely be an extremely energy-intensive process to locate the specific and desired cysteine residue for each protein amongst the milieu of proteins inside the periplasm that are awaiting oxidation and export to the OM, and may even require a range of chaperone proteins. The hypothesis proposed herein is based on the previously published homology model [18] and low-resolution (~4 Å) crystal structure of rMOMP [39], suggesting that COMC proteins possess cysteine-rich regions or ‘pockets’ that act as general target regions for intermolecular disulphide bond formation. The exact interactions between the proteins comprising the COMC have yet to be established, and predictions based on relative protein abundance can be useful in deciphering the likely interacting proteins. Since MOMP is the most abundant of the COMC proteins [15], it seems highly likely that much of MOMP’s intermolecular disulphide bonding will occur with other MOMPs with the potential for some additional cross-linking to other COMC proteins to form an extensive network within the protein mesh.

Using our homology model [18] in conjunction with the low-resolution crystal structure for rMOMP [39], seven key cysteine mutants were created based on their locations within the β-barrel. An obvious cysteine-rich pocket exists in the region of C136, C201 and C203, highlighted in Figure 1A. These residues are unlikely to form intramolecular disulphide bonds due to their close proximity and steric hindrance; instead, they should be available to form intermolecular bonds with adjacent cysteine-rich proteins. Furthermore, adopting the cysteine-rich pocket hypothesis, as C226 is located on the same face at the top of the barrel, this residue was also predicted to be involved in intermolecular disulphide bonding. Due to MOMP’s abundance, and thus its likely role as an interacting partner, it is reasonable to expect that MOMP can form such interactions with the same residues on neighbouring MOMPs. Neutralisation of these residues through mutation to alanine, resulting in truncation of the side chain whilst maintaining the hydrophobicity of the region, was thus pursued in order to investigate the clustering behaviour of MOMP within the *E. coli* outer membrane. Mutation to serine was avoided to prevent salt bridge or hydrogen bond formation due to serine’s associated polarity.

Comparable expression within the membrane of *E. coli* was observed between all seven mutants and wild-type rMOMP with Western blotting of the outer membrane fraction (Figure 1B). Outer membranes were prepared with an initial sarkosyl solubilisation step to remove inner membranes [36] followed by an outer membrane solubilisation step with SB3-14. During membrane preparation, an additional centrifugation step at 16,740× *g* following cell lysis was included in order to remove inclusion bodies from this analysis, thus ensuring the analysis of membrane-associated rMOMP. Additionally, rMOMP of *C. trachomatis*, also induced with 1 mM IPTG and expressed at 25 °C for less than 3 h to encourage correct folding and export, has previously been shown to localise to the *E. coli* outer membrane with whole-cell immunoblots, suggesting that *E. coli*’s protein folding and export machinery are sufficient for rMOMP membrane localisation [23]. Thus, these results are consistent with the absence of a detrimental effect of any combination of the mutations on rMOMP stability and export.

### 3.2. Epifluorescence and TIRF Imaging Demonstrated Differences in Membrane Distribution of rMOMP Mutants

OmpA is a well-characterised and abundant *E. coli* outer membrane protein and, therefore, was labelled with an anti-OmpA antibody coupled with an Alexa Fluor 488 secondary antibody in order to demonstrate the membrane localisation of a homogeneously distributed protein. Importantly, OmpA does not form intermolecular disulphide bonds, as predicted with MOMP.

Initially, mutants C201A, C203A and C201/3A were analysed using epifluorescence and TIRF imaging in order to obtain preliminary low-resolution images of whole *E. coli* cells. It is recommended that a sample concentration of ~4 × 10^8^ cells mL^−1^ (equating to an OD_600_ of 0.5) is used for a desirable distribution of cells. From Figure 2A, it is indisputable that OmpA forms a distinct and solid ring when imaged by epifluorescence and presents as a solid block of fluorescence with TIRF, signifying a homogenous distribution throughout the membrane. However, wild-type rMOMP exhibits cells with a clustered pattern of fluorescence observed in both microscopy techniques (Figure 2B), likely due to intermolecular disulphide bond formation, which prevents the even distribution of rMOMP in the outer membrane. It was hypothesised that the mutation of key cysteine residues, considered important for intermolecular disulphide bonding, would result in a more homogenous membrane localisation pattern, as observed for OmpA. Figure 2C–E represent the single C201A, C203A and double C201/3A mutants, respectively. Mutants C201A and C201/3A appeared to exhibit a more homogenous distribution (Figure 2C, E, respectively) compared to the single C203A mutant (Figure 2D), which still appeared to show a high degree of clustering.

### 3.3. Quantitative Analysis of Direct Stochastic Optical Reconstruction Microscopy (dSTORM)

The qualitative nature of epifluorescence imaging, whereby a single snapshot of a group of cells is presented, is limited as a measure of clustering due to the subjectivity of visual interpretation. Therefore, dSTORM was used to obtain both higher-resolution images and, more importantly, quantitative data from a greater number of cells for a more detailed analysis of MOMP clustering. Eight-well No 1.5H glass-bottom Ibidi slides were utilised to enable high-throughput sample preparation and imaging, in addition to improved cell labelling compared to alternative methods and elimination of buffer evaporation during imaging.

C-terminal His-tagged rMOMP was labelled with an Alexa Fluor 647 anti-His antibody, an optimal fluorophore for super-resolution microscopy imaging due to its high photon numbers, low duty cycles, high survival fractions and many switching cycles [40], permitting the capture of large datasets of localisations before photobleaching. For each sample, at least ten dSTORM datasets were collected in order to obtain a minimum of fifty regions of interest (ROIs) per sample, where an ROI is treated as one cell. In total, between 51 and 94 cells were collected per dataset (Table 1). Samples were treated with 3% (*w*/*v*) BSA to minimise nonspecific antibody binding with control cells imaged with and without labels to identify autofluorescence and nonspecific interactions, respectively. Samples were also treated with lysozyme in order to improve the accessibility of the fluorophore to the C-terminal His-tag on rMOMP, which is located at the periplasmic side of the membrane (Figure 1A). Figure 3 illustrates a selection of the super-resolution dSTORM images for control OmpA, wild-type rMOMP and all seven of the cysteine rMOMP mutants, each of which is composed of a dataset of single-molecule localisations. The localisation precision histograms are displayed in Appendix A. Datasets were first drift -corrected to account for minuscule sample holder movements and grouped to merge trails before quantitative analysis of clustering was conducted using ClusDoC software [38], which implements the density-based spatial clustering of applications with noise (DBSCAN) algorithm [41]. Each bacterial cell was treated as an ROI and the radius for detection was set as median localisation precision, which in this instance was 20 nm, and a cluster defined as a minimum of three points.

### 3.4. Significant Decreases in Clustering Were Observed Between Wild-Type rMOMP and Cysteine Mutants C201/203A, C136/201A, and C226A

The relative cluster density is a measure of the local variance of molecular density within the defined clusters. Analysis of the quantitative clustering data revealed that wild-type rMOMP was the most clustered sample, with an average relative density of clusters of approximately 3.7 arbitrary units (AU) compared to the OmpA control, representing homogenous dispersion and low clustering, which had an average density of 2 AU. Both datasets support the observations made in the preliminary epifluorescence and TIRF microscopy experiments. The mutants with the most significant decrease in clustering were the two double mutants C201/203A and C136/201A and single mutant C226A with average densities between 1.9 AU and 2.1 AU (Table 1 and Figure 4A).

A one-way analysis of variance (ANOVA) (F(8,630) = 35.8, *p* < 0.0001, Figure 4B) against the null hypothesis that there is no statistically significant difference between groups permitted the rejection of the null hypothesis with a *p*-value less than the 0.05 alpha value, in addition to an F critical value (1.95) lower than the value of F. In order to determine where these differences occurred, for groups with unequal sample sizes, a Tukey post-hoc test was conducted (Table 2). From this test, it was apparent that clustering differences between OmpA and wild-type rMOMP were statistically significant, as well as those between wild-type rMOMP and all seven of the cysteine mutants.

A statistically significant difference was observed between OmpA and the following mutants: C201A, C203A, C136A and C136/203A (Table 2). This suggests that whilst these mutants exhibited reduced clustering from wild-type rMOMP, they were not as homogenous as the control group OmpA. On the contrary, no statistically significant difference was observed between OmpA and mutants C201/203A, C136/201A and C226A, (Table 2, highlighted in grey), suggesting that clustering in these mutants had been reduced to exhibit a homogenous dispersion similar to that observed with the OmpA control.

### 3.5. A Compensatory Mechanism of Disulphide Bond Formation between Clusters

With regard to the double mutants, in most instances, their respective single mutants did not exhibit the same decrease in clustering, indicating that two mutations in combination were required to produce a more significant response. Contrary to this, the double mutant C136/203A did not decrease clustering to the same extent as that observed for the other two double mutants in the pocket. Comparison with the other mutant data suggested that residue C201, the remaining cysteine in this pocket, has a more critical role in disulphide bond formation than the neighbouring C203. However, the effect of the C201A single mutant was less significant than that of the C203A single mutant. This can be rationalised through our proposal that the COMC proteins target cysteine-rich pockets during disulphide bond formation as opposed to specific residues. Analysis of the putative location of these cysteines in the MOMP homology model (18) suggests that residue C342, on the opposite side of the barrel, is most likely interacting with the cysteine-rich pocket (C136, C201 and C203) investigated (Figure 5). Based on the results obtained, when only one cysteine residue within the C136, C201 and C203 pocket is mutated, the two remaining cysteine residues can compensate and continue to form the disulphide bond. C201 appears to be much more effective at forming the disulphide bond alone when C136 and C203 are mutated in comparison to either C136 or C203 when singularly available. This may be because residue C201 is situated higher within the β barrel and in a more sterically favourable position to interact with C342, although the neighbouring residues are still available to form a compensatory disulphide bond if required. Hence, a compensatory mechanism can be inferred; otherwise, it would be expected that the mutation of any one particular residue would cause a decrease in both single and double mutants in equal (or greater) quantities.

### 3.6. Residue C226 May Play a Key Role in Intermolecular Disulphide Bonding with Pocket C49–52–56

Mutant C226A exhibited a significant effect on clustering despite being a single mutation that is not arranged in a cluster. Again, with respect to cysteine residue locations within the β-barrel, these data suggest that this residue is key in forming disulphide bonds with a cysteine-rich region formed by residues C49, C52 and C56. Due to its alignment with C226, it is hypothesised that residue C52 would form the most favourable interaction, with its flanking residues C49 and C56 available for compensation (Figure 5), as required.

## 4. Discussion

Chlamydial infections are responsible for a range of human diseases as well as being linked to a number of secondary inflammatory conditions such as atherosclerosis and reactive arthritis. The investigation of chlamydial biology to understand its mechanisms of infection has been complicated by its biphasic lifecycle, whereby infectious EBs circulate the host before differentiating into the metabolically active RBs. In the absence of detectable peptidoglycan, EBs employ the unique COMC to provide cellular structural stability, and whilst the COMC protein composition has been elucidated, the disulphide bonding mechanism, membrane localisation, and interactions remain to be characterised. MOMP, which accounts for the majority of the cysteine-rich membrane proteins within the COMC, is also upregulated during the EB stage of the lifecycle, both of which are suggestive of a key role in cell structural stability [13]. To date, the only experimental data characterising potential disulphide bonding in MOMP was conducted in *C. trachomatis* MOMP. These data were obtained from peptide fragments that could have reassociated in a number of ways, which in fact have given rise to conflicting results [21,22]. Our *C. pneumoniae* MOMP homology model [18], supported by a low-resolution crystal structure of rMOMP [39], enabled the rational design of key cysteine mutants and the emergence of a hypothesis, developed during the course of this work, which suggests that due to the high number of COMC proteins and consequently the numerous cysteine residues available for disulphide formation, the COMC proteins target cysteine-rich pockets as opposed to specific cysteine residues. This is also supported by analysis of the β-barrel structure, which revealed cysteine-dense regions referred to as cysteine-rich pockets that were hypothesised to be involved in intermolecular disulphide bonding with highly abundant MOMP, but also with the potential to interact with other cysteine-rich outer membrane proteins to form a strongly stabilised mesh.

The biphasic lifecycle of *Chlamydia*, in addition to its requirement for a host cell to survive, has made genetic manipulation complex and problematic, and therefore this research was conducted recombinantly in *E. coli* where rMOMP was previously shown to localise at the *E. coli* outer membrane through immunoblotting analysis under similar expression conditions [23]. Not only do *Chlamydia*’s own Dsb homologues share 20% identity to those of *E. coli* but disulphide bonding has also been shown to occur in *E. coli* under aerobic conditions in the absence of its Dsb proteins [26], suggesting that such bonding is able to form somewhat spontaneously and thus *E. coli* is a promising host for the study of MOMP disulphide bonding.

A key cysteine-rich pocket includes residues C136, C201 and C203, which were neutralised through mutation to alanine along with a potential interactor C342 located on the opposite side of the β-barrel, in addition to C226, which was found nearer the extracellular side of the β-barrel. From low-resolution imaging with epifluorescence microscopy, it was apparent that wild-type rMOMP was forming clusters within the *E. coli* membrane, characterised as speckled fluorescence. This was also observed in previous research into the membrane localisation of *C. trachomatis* MOMP when expressed recombinantly in *E. coli* [23]. However, due to the qualitative nature of epifluorescence microscopy imaging, and thus the subjectivity associated with the analysis of such visual data, a higher-resolution technique was pursued to enable detailed quantitative characterisation of these differences.

High-resolution dSTORM data revealed wild-type rMOMP to be highly clustered and showed the OmpA control as more homogeneous, whereby a cluster is defined as three or more interactions. Double mutants C201/203A and C136/201A reduced the clustering of rMOMP most significantly, suggesting that within cysteine-rich regions, a compensatory mechanism is occurring whereby neighbouring cysteine residues can continue to form intermolecular disulphide bonds in the absence of the most important cysteine residue. Notably, the single mutant C226A produced a significant effect on disulphide bonding. It is likely that this residue is interacting with an additional cysteine-rich region formed by residues C49, C52 and C56 at the top of the barrel on the opposing face. It is also important to note that whilst C203A appeared to be more clustered than wild-type MOMP in the preliminary epifluorescence and TIRF images, the same mutant demonstrated significantly reduced clustering in comparison to the wild type following statistical analysis of the dSTORM data. Whilst epifluorescence and TIRF microscopy are excellent preliminary experiments for the visualisation of proteins at the bacterial membrane, this discrepancy again highlights that epifluorescence and TIRF are limited in that they produce a qualitative snapshot of a small number of cells, as opposed to dSTORM data analysis whereby a minimum of fifty cells were statistically analysed, again, emphasising the importance of implementing robust quantitative high-resolution microscopy techniques.

These results support the notion that, during COMC formation, a general cysteine-rich region is targeted, as opposed to specific residues, although it is possible that some residues within these regions, such as C201, have more substantial roles, perhaps even depending on the specific target protein partner in the COMC. However, until further structural data detailing the location of the cysteine residues within these other partner proteins of the COMC emerge, characterisation of these interactions remains extremely challenging. Understanding both the COMC architecture and the disulphide bonding within key proteins in greater depth can lead to the development of novel therapeutics, through weakening of the infectious chlamydial EBs as they circulate the host.

## 5. Conclusions

MOMP is a major component of the COMC, the protective disulphide-linked protein mesh that surrounds the outer membrane of Chlamydial EBs. Despite having an essential role in cell structural stability and survival within the host, little is known about the exact interactions that occur between the multitude of proteins present in this network. Our research identified cysteine rich regions within MOMP’s tertiary structure and introduced a series of mutations in order to assess their effect upon membrane clustering, assessed with high-resolution dSTORM. Our results indicated that single cysteine mutants positioned within the pocket did not significantly affect disulphide bond formation, suggestive of a compensatory mechanism whereby neighbouring cysteine residues are able to form the bond instead. Additionally, two lone cysteine residues positioned opposite the rich pocket regions were identified to have key roles in disulphide bonding, indicating two potential interfaces for interaction on the MOMP β-barrel. As a result, we proposed a cysteine rich pocket hypothesis for disulphide bond formation in *Chlamydia*, whereby a general cysteine-rich region is targeted as opposed to specific residues in order to quickly and effectively form the protective cysteine rich COMC. 

## Figures and Tables

**Figure 1 biology-09-00344-f001:**
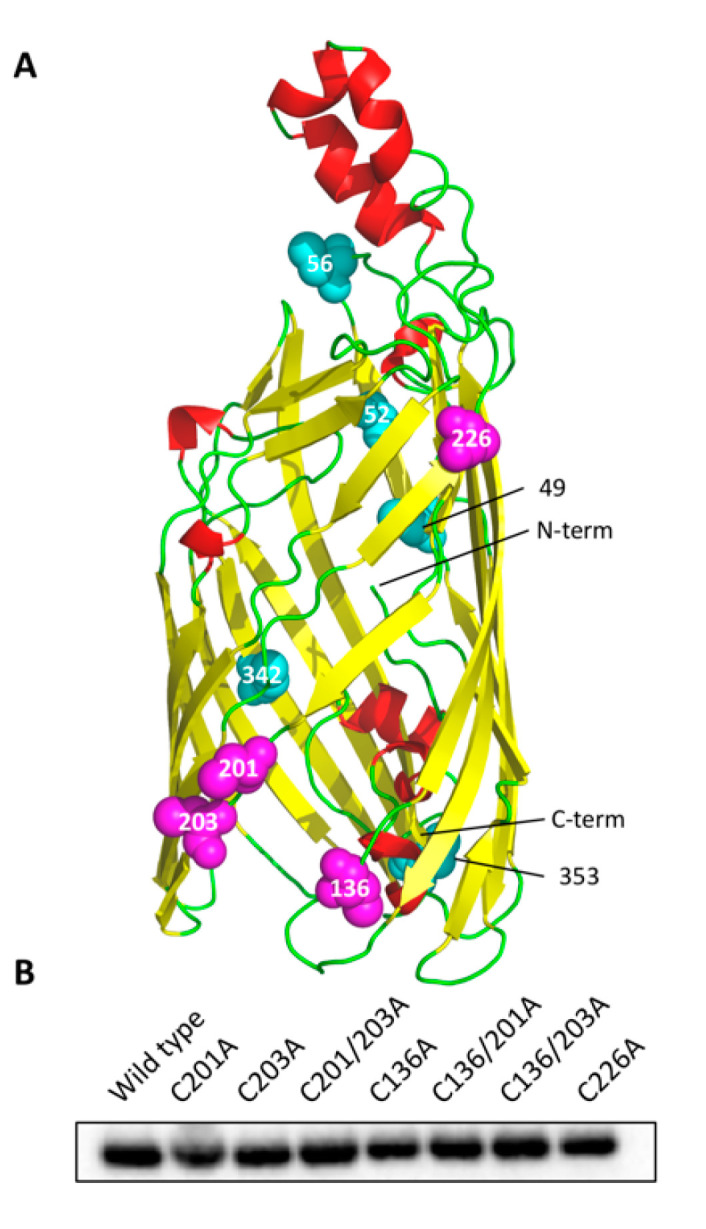
(**A**) *C. pneumoniae* major outer membrane protein (MOMP) homology model with cysteine residues highlighted. The four cysteines of interest (C136, C201, C203 and C226) are labelled and shown as magenta spheres with the remaining cysteines shown as cyan spheres. The N-terminus is located on a loop inside the barrel and the C-terminus (His-tagged) on a β-sheet on the periplasmic side of the membrane. (**B**) Western blot of cysteine-mutated recombinant MOMP (rMOMP) expressed in *E. coli* outer membranes. Protein was detected using a 6×His epitope tag monoclonal mouse IgG primary antibody and an anti-mouse IgG (H+L)-HRP-conjugated secondary antibody.

**Figure 2 biology-09-00344-f002:**
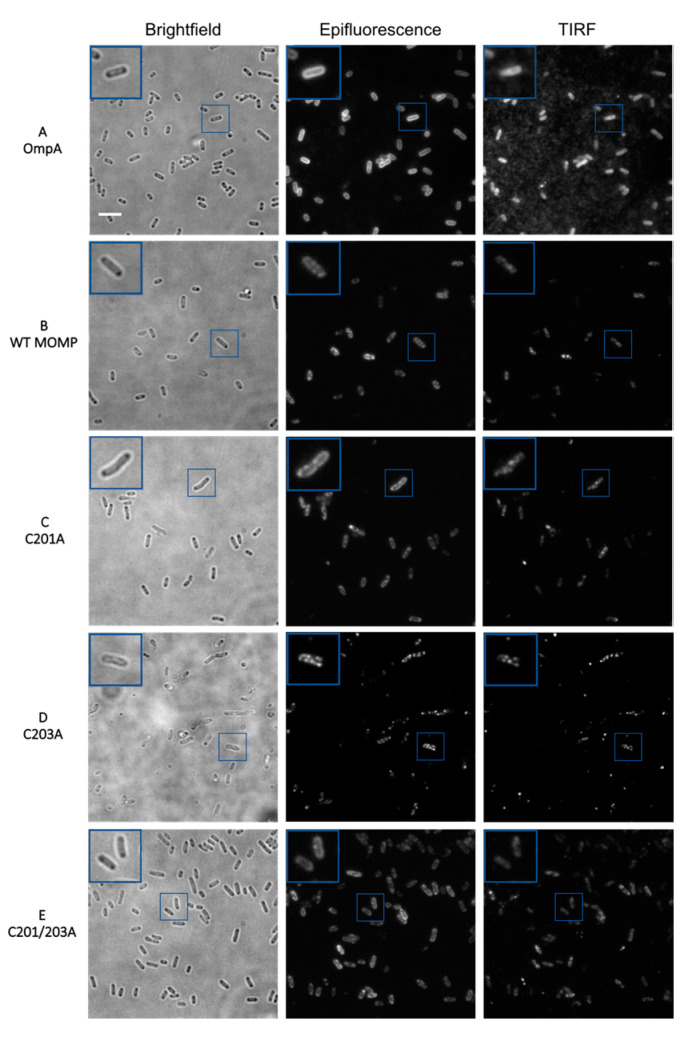
Brightfield, epifluorescence and total internal reflection fluorescence (TIRF) microscopy images of rMOMP cysteine mutants in C41 (DE3) *E. coli* cells. (**A**) *E. coli* anti-OmpA; (**B**) wild-type rMOMP; (**C**) rMOMP C201A; (**D**) rMOMP C203A; (**E**) rMOMP C201/3A. rMOMP-expressing cells were stained using anti-His rabbit primary antibody coupled with anti-rabbit Alexa Fluor 488 secondary antibody, with control *E. coli* cells stained using anti-OmpA rabbit primary antibody coupled with anti-rabbit Alexa Fluor 488 secondary antibody. Scale bar is 5 µm, inset is ×2 magnification.

**Figure 3 biology-09-00344-f003:**
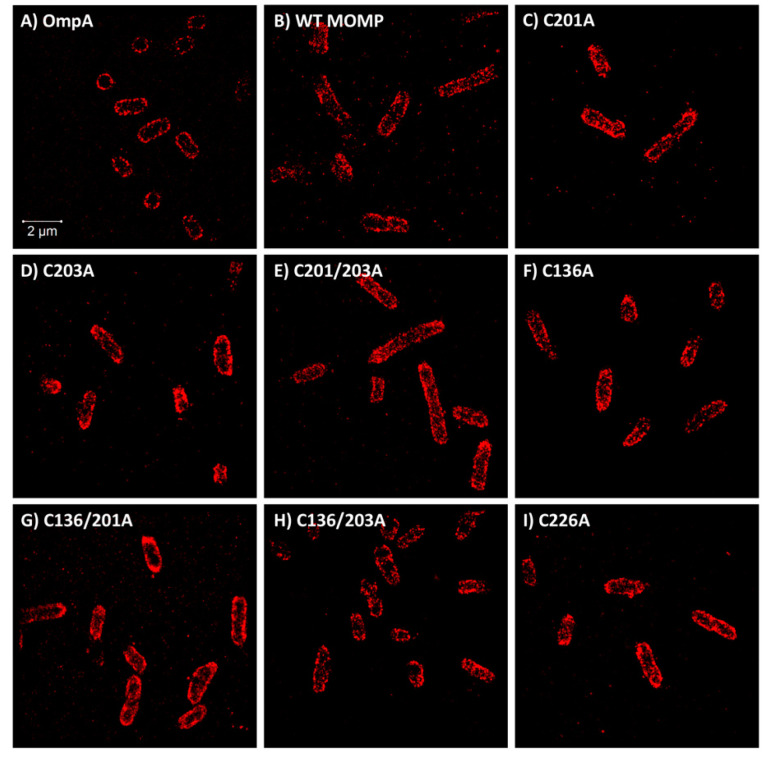
Super-resolution dSTORM images of *E. coli* cells. (**A**) *E. coli* anti-OmpA; (**B**) wild-type rMOMP; (**C**) rMOMP C201A; (**D**) rMOMP C203A; (**E**) rMOMP C201/203A; (**F**) rMOMP C136A; (**G**) rMOMP C136/201A; (**H**) rMOMP C136/203A; and (**I**) rMOMP C226A. All cells were induced for a total of 3 h, with rMOMP-expressing cells labelled with an anti-His Alexa Fluor 647 antibody and OmpA first with anti-OmpA primary antibody coupled with an anti-647 secondary antibody. Localisation precision histograms are provided in Appendix A.

**Figure 4 biology-09-00344-f004:**
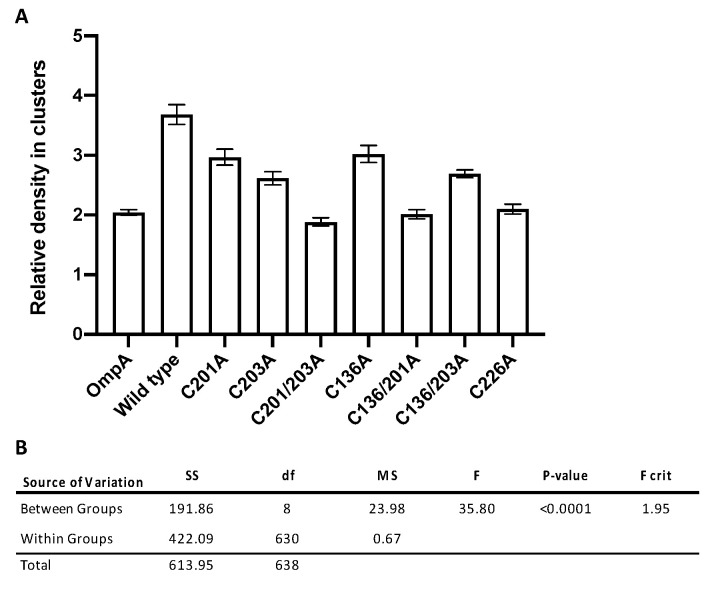
(**A**) Graph displaying the relative cluster densities of OmpA, wild-type rMOMP, and rMOMP cysteine mutants. dSTORM localisation data were analysed with ClusDoC software before the mean and standard error were calculated for each sample (Table 1). All data were collected and processed under the same conditions. The relative density in clusters is provided as a standardised arbitrary unit and error bars indicate the standard error. (**B**) One-way ANOVA statistical analysis of dSTORM localisation data. The null hypothesis that there is no statistically significant difference between groups can be rejected as *p* < 0.05 and the F critical value of 1.95 is less than F at 35.80. SS, sum-of-squares; df, degrees of freedom; and MS, mean squares.

**Figure 5 biology-09-00344-f005:**
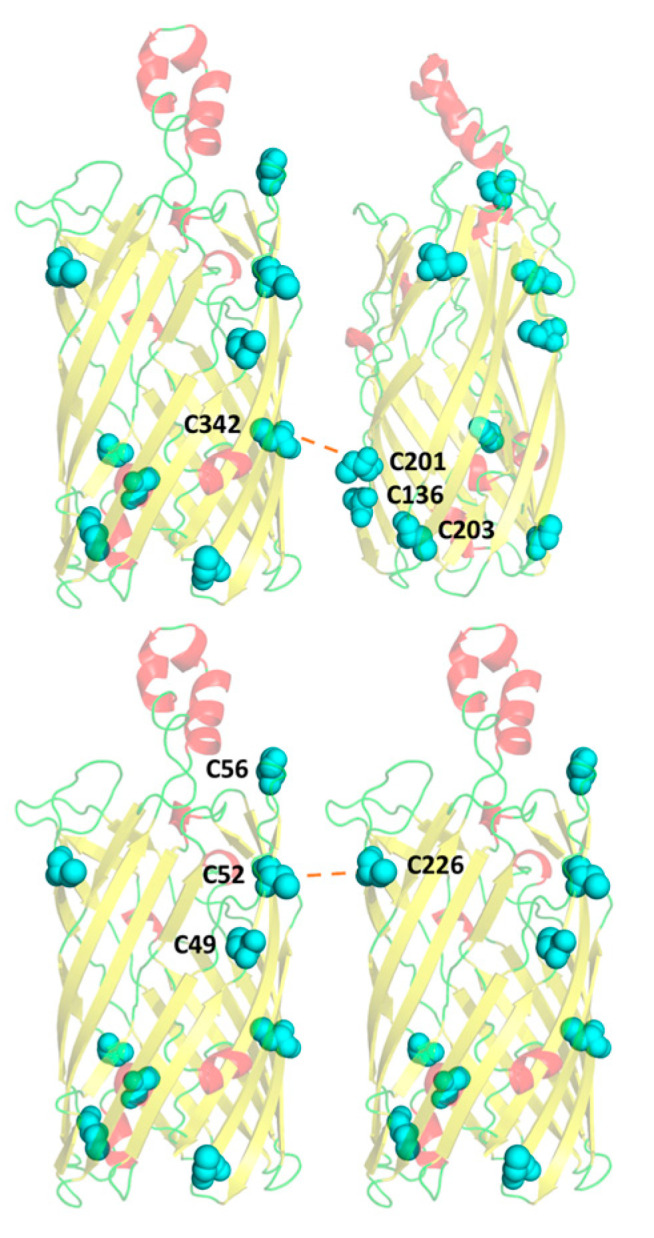
Two MOMP monomers indicating the hypothesised intermolecular disulphide bonds between C342 and C201, and C52 and C226. The nine cysteines residues are shown in cyan, with key residues labelled. Hypothesised intermolecular disulphide bonding is indicated by red dashed lines.

**Table 1 biology-09-00344-t001:** The relative cluster densities and associated standard errors of OmpA, wild-type rMOMP, and rMOMP cysteine mutants. Direct stochastic optical reconstruction microscopy (dSTORM) localisation data were analysed with ClusDoC software before the mean and standard error were calculated for each sample. All data were collected and processed under the same conditions. SE, standard error; N, number of cells analysed.

Construct	Relative Density in Clusters	SE	n
OmpA	2.04	0.05	94
Wild type	3.68	0.17	59
C201A	2.97	0.13	51
C203A	2.61	0.11	62
C201/203A	1.88	0.07	71
C136A	3.02	0.14	69
C136/201A	2.01	0.08	78
C136/203A	2.69	0.07	90
C226A	2.10	0.08	65

**Table 2 biology-09-00344-t002:** Tukey post-hoc test to indicate where differences between groups arise. Following the rejection of the null hypothesis with a one-way ANOVA, a Tukey post-hoc test for groups of different sizes was conducted. Clustering between OmpA and wild-type rMOMP was deemed to be significantly different, as well as clustering between wild-type rMOMP and every cysteine mutant, as assessed with q-crit values above 4.39, the *q*-value derived from a studentised *q* table with alpha as 0.05. Mean difference refers to the difference in clustering values in arbitrary units (AU); N to the number of cells analysed; SE, standard error. The mutants with the most significant decrease in clustering are highlighted in grey.

Important Pairs		Mean Difference	N (Group 1)	N (Group 2)	SE	q-Crit	Significant
OmpA	Wild type	1.64	94	59	0.10	17.05	YES
Wild type	C201A	0.71	59	51	0.11	6.44	YES
Wild type	C203A	1.07	59	62	0.11	10.15	YES
Wild type	C201/203A	1.80	59	71	0.10	17.63	YES
Wild type	C136A	0.66	59	69	0.10	6.45	YES
Wild type	C136/201A	1.67	59	78	0.10	16.71	YES
Wild type	C136/203A	0.99	59	90	0.10	10.21	YES
Wild type	C226A	1.58	59	65	0.10	15.21	YES
OmpA	C201A	0.93	94	51	0.10	9.20	YES
OmpA	C203A	0.57	94	62	0.09	6.03	YES
OmpA	C201/203A	0.16	94	71	0.09	1.75	NO
OmpA	C136A	0.98	94	69	0.09	10.65	YES
OmpA	C136/201A	0.03	94	78	0.09	0.33	NO
OmpA	C136/203A	0.65	94	90	0.09	7.60	YES
OmpA	C226A	0.06	94	65	0.09	0.60	NO

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
