# Peer review of "Super-Resolution Fluorescence Microscopy Reveals Clustering Behaviour of Chlamydia pneumoniae’s Major Outer Membrane Protein"

_biology, 2020, doi:10.3390/biology9100344_

Round 1
Reviewer 1 Report
In the submitted manuscript by A. E. Danson et. al. entitled “Super-resolution fluorescence microscopy reveals clustering behavior of Chlamydia pneumoniae´s major outer membrane protein” the authors investigated the clustering behavior of the major outer membrane protein by immunoflourescence and high resolution microscopy. The manuscript is well written and presenting and discussing the findings are clearly addressed. Therefore, I do believe that the manuscript deserves to be published in Biology in its current shape. I do have one question to the authors. By using epifluorescence and TIRF microscopy the authors showed that the highest degree of clustering is for the mutant C203A followed by the wild type while dSTORM suggests the opposite scenario. Why this discrepancy?
Author Response
Point 1:
By using epifluorescence and TIRF microscopy the authors showed that the highest degree of clustering is for the mutant C203A followed by the wild type while dSTORM suggests the opposite scenario. Why this discrepancy?
Response 1:
Thank you for taking the time to review our research article. We agree with your point regarding mutant C203A and the discrepancy between TIRF/EPI images and the STORM data and we have included additional information within the manuscript to clarify the observed differences. Whilst epifluorescence and TIRF are excellent preliminary experiments for the visualisation of proteins at the bacterial membrane, they are limited in that they represent qualitative data and additionally are a single snapshot of one particular group of cells, and are thus subject to a level of subjectivity. Due to this, STORM was implemented in order to capture data from a minimum of 50 individual cells, with the data statistically analysed to remove any possible subjectivity and in turn provide a more robust analysis of clustering.
Reviewer 2 Report
Overall, this review is very thorough and does provide greater insight into testing their hypothesis using several different techniques. The statistical analysis is there and in great detail, and the results are explained clearly.
There are only a few points that would be strongly recommended prior to publication, all quite minor revisions if possible.
Comment 1 [Minor]: Other Recent Studies? I understand that this paper focuses on specific studies done by the group and is highly specific. However, additional studies and related studies should be brought in as well to add more scope and breadth to each section, specifically the discussion. While searching online, I have found some more recent studies that touch on MOMP, cysteine abundancy, as well as fluorescence microscopy studies related to these – can you double check these and make sure you are not missing any? Out of the ~40 references listed, only 8 of them are within the past 4 years; this is a bit concerning to me regarding the timeliness of the matter. It would be nice if a few more recent papers are added in if present.
Comment 2 [Minor]: Longer Discussion. The discussion is well-written, but I believe it should be longer and a bit more detailed, especially when it comes to comparing your results to others. I trust that if Comment #1 is covered, then Comment #2 will follow along with the additional studies that can be referred to, to target the relevancy of the paper. If there are no additional studies to date from Comment #1, then this can be ignored because it currently reads well.
Comment 3 [Minor]: Review reference formatting. Some years are bolded, while others are not. Very minor, but please make them consistent!
Author Response
Comment 1 [Minor]: Other Recent Studies? I understand that this paper focuses on specific studies done by the group and is highly specific. However, additional studies and related studies should be brought in as well to add more scope and breadth to each section, specifically the discussion. While searching online, I have found some more recent studies that touch on MOMP, cysteine abundancy, as well as fluorescence microscopy studies related to these – can you double check these and make sure you are not missing any? Out of the ~40 references listed, only 8 of them are within the past 4 years; this is a bit concerning to me regarding the timeliness of the matter. It would be nice if a few more recent papers are added in if present.
Comment 2 [Minor]: Longer Discussion. The discussion is well-written, but I believe it should be longer and a bit more detailed, especially when it comes to comparing your results to others. I trust that if Comment #1 is covered, then Comment #2 will follow along with the additional studies that can be referred to, to target the relevancy of the paper. If there are no additional studies to date from Comment #1, then this can be ignored because it currently reads well.
Response 1 and 2:
Thank you for taking the time to act as referee to our original research article. The main focus on Chlamydial MOMP within the literature is its role as a vaccine candidate, and thus the multitude of recent papers are heavily focussed on vaccinology and the associated immunological response. As our manuscript is intended as an original research article and not a review article, there is not scope in this manuscript to cover this additional field of research. Furthermore, the less well studied field of recombinant and structural studies is slow paced due to the difficulties of both Chlamydial genetic manipulation as well as membrane protein expression and purification, and therefore there is limited research regarding the analysis of disulphide bonding, which is in fact the main focus of our manuscript. The only potentially relevant recent paper is Hepler et al., Protein. Sci. 2018 (https://onlinelibrary.wiley.com/doi/full/10.1002/pro.3501) but here the authors only refer to cysteines in the same way as we have in our manuscript i.e. referenced from the older papers. Importantly, there was no original research reported in the paper of Hepler with regard to the cysteine residues, which also is why it hasn’t been included here, whereas we have included the original relevant research publications.
Comment 3 [Minor]: Review reference formatting. Some years are bolded, while others are not. Very minor, but please make them consistent!
Response 3:
All references have been updated to a consistent format.